# Maintaining and breaking symmetry in homomeric coiled-coil assemblies

Guto G. Rhys [1], Christopher W. Wood [1], Eric J.M. Lang [1,2], Adrian J. Mulholland[1,2,3], R. Leo Brady [4], Andrew R. Thomson [1,5] & Derek N. Woolfson [1,2,4]

In coiled-coil (CC) protein structures α-helices wrap around one another to form rope-like assemblies. Most natural and designed CCs have two–four helices and cyclic ($C_n$) or dihedral ($D_n$) symmetry. Increasingly, CCs with five or more helices are being reported. A subset of these higher-order CCs is of interest as they have accessible central channels that can be functionalised; they are α-helical barrels. These extended cavities are surprising given the drive to maximise buried hydrophobic surfaces during protein folding and assembly in water. Here, we show that α-helical barrels can be maintained by the strategic placement of β-branched aliphatic residues lining the lumen. Otherwise, the structures collapse or adjust to give more-complex multi-helix assemblies without $C_n$ or $D_n$ symmetry. Nonetheless, the structural hallmark of CCs—namely, knobs-into-holes packing of side chains between helices—is maintained leading to classes of CCs hitherto unobserved in nature or accessed by design.

[1] School of Chemistry, University of Bristol, Cantock's Close, Bristol BS8 1TS, UK. [2] BrisSynBio, University of Bristol, Life Sciences Building, Tyndall Avenue, Bristol BS8 1TQ, UK. [3] Centre for Computational Chemistry, School of Chemistry, University of Bristol, Cantock's Close, Bristol BS8 1TS, UK. [4] School of Biochemistry, University of Bristol, Medical Sciences Building, University Walk, Bristol BS8 1TD, UK. [5] School of Chemistry, University of Glasgow, Glasgow G12 8QQ, UK. Correspondence and requests for materials should be addressed to G.G.R. (email: Guto.Rhys@bristol.ac.uk) or to D.N.W. (email: D.N.Woolfson@bristol.ac.uk)

α-Helical coiled coils (CCs) were among the first natural protein structures envisaged[1]. Now, they are estimated to account for 5–10% of all protein-encoding sequences across all genomes[2]. The classical view of CC sequences and structures is as follows.

CCs are quaternary structures comprising two or more α-helices supercoiled around each other with cyclic ($C_n$) or dihedral ($D_n$) symmetry. The simplest CCs are parallel ($C_2$) and anti-parallel ($D_1$) dimers (Fig. 1a, b). Invariably for water-soluble CCs, the helices are amphipathic and assemble to remove their hydrophobic faces from water. The sequence signature of canonical CCs is a 7-residue (heptad) repeat of hydrophobic (h) and polar (p) residues, hpphppp, often denoted abcdefg. As the 3,4 spacing of hydrophobic residues does not quite match the 3.6-residues per turn of the α-helix, the helices wrap around each other with a left-handed supercoil (Fig. 1a, b).

Residues at the helical interfaces pack with knobs-into-holes (KIH) interactions[1,3–5], which is distinct from the packing of helices in proteins generally[6]. In KIH packing, a knob side-chain from one helix docks into a diamond-shaped hole of four side chains on a neighbouring helix, Fig. 1c. KIH packing is the foundation of Crick's CC postulate[1], from which CC sequence, structure and symmetry follow. Thus, while potential CCs can be predicted from sequences[7], they are only confirmed by KIH interactions in 3D structures using programmes like SOCKET[5]. This has been used to find CCs in the RCSB Protein Data Bank leading to the CC+ database[8] and a Periodic Table of Coiled Coils[9]. These reveal that the vast majority of the current CC structures are dimers, trimers or tetramers (937/1012 (93%) of the CCs with 50% sequence identity or less); and, by inspection, these assemblies predominantly have $C_n$ or $D_n$ symmetry.

Sequence variations in the heptad repeats discriminate parallel dimers, trimers and tetramers:[10] a = Ile plus d = Leu, directs dimer; a = d = Ile, trimer; and a = Leu plus d = Ile, tetramer. These preferences reflect differences in KIH packing between oligomers arising from how the Cα–Cβ bond vector of the knob engages with the hole. This is through one of three main packing

arrangements—parallel, perpendicular and acute[5,10]—and variations on this[11] (Fig. 1d–g). Thus, parallel CC dimers, trimers and tetramers are extremely well understood and can be modelled or built reliably de novo[12–14].

That is the classical view of CCs, and we refer to these as canonical heptad-based CCs (Fig. 1d)[15].

Structures of natural, engineered and de novo CC assemblies above tetramer are being resolved, including: water-soluble and membrane-spanning CC pentamers;[16–18] a 10-helix, viral DNA-piloting tube;[19] larger membrane-active pores;[20–22] water-soluble CC pentamers–heptamers engineered serendipitously[23,24] or designed computationally;[25,26] and beyond these, there are some more-complex CCs[9,15]. Many of these have central channels or pores; they are α-helical barrels rather than simpler α-helical bundles[27]. Such barrels are of interest because of their channel/pore functions or potential for functionalization. However, they challenge the primacy of hydrophobicity in determining CC structures, raising the question: what sequences maintain barrels and oppose complete hydrophobic collapse?

Understanding higher-order CCs requires an expansion of the traditional CC concept[28–30]. The helical interfaces extend past the a and d residues, with e and g sites participating in KIH interactions[15]. Residues at these sites do not have to be hydrophobic, though it is convenient to think in these terms. The canonical heptad repeat, hpphppp, can be embellished in three ways: hpphpph or hpphhpp; hpphhph; and hphhphp or hhphphp, referred to as Type 1, 2 and 3 sequences, respectively, (Fig. 1d–f)[15]. These sequence types promote different CC assemblies. For example at the extremes: dimeric CCs are Type-N; whereas, Type-3 sequences set-up two distinct hydrophobic seams on each helix leading to large barrels[28,30], which have been exploited to make tubular materials from de novo peptides[31].

Type-2 sequences can lead to α-helical barrels with 5–7 helices, including: the aforementioned pentamers; the serendipitously discovered hexamer, CC-Hex[24] and slipped heptameric e = g = Ala permutant of the GCN4 leucine zipper, GCN4-pAA[23]. In these structures, two seams are presented on a single α-helix

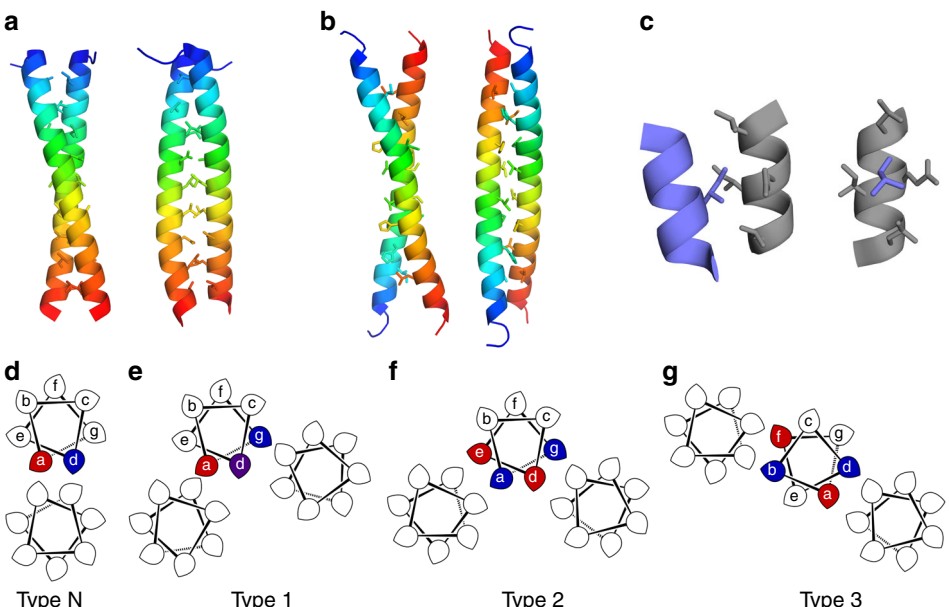

**Fig. 1** Coiled-coil structure, symmetry and sequence. **a**, **b** Parallel ($C_2$ symmetric; PDB entry 4DZM) and antiparallel ($D_1$ symmetric; 1HF9) CC dimers. **c** Knobs-into-holes (KIH) interaction with an isoleucine knob packing into a diamond-shaped hole (grey). **d–g** Helical wheels for classical Type-N (**d**), and for Type-1 (**e**), Type-2 (**f**) and Type-3 interfaces (**g**). All are viewed along the α-helices from the N to C termini, labels are for the canonical a–g nomenclature, and the teardrop shapes indicate the direction of Cα–Cβ bonds. Coloured residues highlight knobs in KIH interactions with parallel (red), perpendicular (blue) and X-layer (purple) KIH types. Unlabelled helices are for reference

**Table 1 Designed peptides and summary of biophysical data**

| Heptad repeat sequence[a] abcdefg | Systematic name suffix[b] | Oligomer state[c] | | DPH binding[d] ($K_D$, μM) | XRD[e] PDB accession codes | XRD oligomer state and structure type |
|---|---|---|---|---|---|---|
| | | SV | SE | | | |
| **ββ** | | | | | | |
| VKEVAfA | -VV | 5.8 | 5.6 | Y (27 ± 10.9) | 6G65 | 6 (barrel) |
| IKEVAfA | -IV | 6.8 | 6.4 | Y (8 ± 2.0) | 6G66 | 7 (barrel) |
| VKEIAfA | -VI | 5.4 | 5.3 | Y (7 ± 1.9) | - | - |
| IKEIAfA | -II | 6.1 | 5.7 | Y (57 ± 9.5) | 6G67 | 8 (barrel) |
| **Lβ** | | | | | | |
| LKEIAfA | -LI | 6.3 | 5.8 | Y (21 ± 3.0) | 4PNA | 7 (barrel) |
| IKEIAfA | -deLI | 6.9 | 6.6 | Y (12 ± 3.0) | 6G6E | 7 (barrel) |
| LKEVAfA | -LV | 6.3 | 5.4 | N | - | - |
| **βL** | | | | | | |
| VKELAfA | -VL | 4.9 | 5.4 | N | - | |
| IKELAfS | -IL-Sg | 6.4 | 6.1 | N | 6G68 | 6 (collapsed) |
| IKELAfS | -IL-Sg-L17E | 5.9 | 5.9 | Y[f] (62 ± 7.2) | 6G69 | 7 (barrel) |
| **LL** | | | | | | |
| LKELAfA | -LL | 6.1 | 5.9 | N | 6G6A | 6 (collapsed) |
| LKELAfA | -LL-L17Q | 6.0 | 6.0 | N | 6G6B | 6 (collapsed) |
| LKELAfA | -LL-L17E | 6.1 | 5.8 | Y[f] (10 ± 6.9) 272 ± 64.9) | 6G6C | 6 (collapsed) |
| **Aromatics (collapsed)** | | | | | | |
| IKEFAfA | -IF | 5.7 | 6.3 | N | - | - |
| FKEIAfA | -FI | 5.7 | 5.7 | - | 6G6F | 6 (collapsed) |
| LKEFAfA | -LF | 5.8 | 5.9 | N | 6G6G | 8 (collapsed) |

[a]Main heptad repeat of each peptide. Amino acids are denoted by standard one-letter code; except l for 4,5-dehydro-leucine (deL). Repeating f positions are occupied from N to C terminus by Q, K, W and Q, respectively. Full sequences are given in Supplementary Table 1
[b]Sequences are described as CC-Type2 with a unique suffix. CC-Type2-VI, CC-Type2-LI and CC-Type2-LV have been described previously as AVKEIA, CC-Hept and ALKEVA, respectively[26]
[c]Oligomer state in solution determined by sedimentation velocity (SV) experiments from a single run fitted to c(s) distributions to 95% confidence limits, or sedimentation equilibrium (SE) experiments ran in triplicate and fitted to a single ideal species. The values are observed molecular weight divided by monomer mass
[d]Yes (Y) or No (N) binding of DPH (1 μM) over a range of peptide concentrations ($n \geq 3$) to give $K_D$ values and standard errors. Dissociation constants are quoted per peptide
[e]Protein crystallography, X-ray diffraction
[f]Binding to 1-(4-trimethylammoniumphenyl)-6-phenyl-1,3,5-hexatriene p-toluenesulfonate (TMA-DPH) a cationic derivative of DPH

allowing interaction with two neighbouring helices (Fig. 1f). These two seams can be considered as two overlaid Type-N interfaces (Fig. 1d). To avoid confusion, a single-heptad repeat is described where one seam has knob residues at a and e and the other knob residues at d and g. Pentameric COMP, CC-Hex and GCN4-pAA have all been engineered to introduce functional groups into the otherwise hydrophobic lumens or to make peptide-based tubular materials[32–35]. Type-2 interfaces have also been exploited in the rational computational design of entirely de novo pentameric, hexameric and heptameric α-helical barrels[25,26,36], which have been engineered to introduce rudimentary binding[34], catalytic[37], and pH-based switch functionalities[36].

Here, we explore the mutability of Type-2 sequences and interfaces through empirical redesigns of the computationally designed heptamer, CC-Hept[26]. Specifically, the a and d sites are mutated en bloc to all combinations of the aliphatic residues Ile, Leu and Val. In solution, these peptides form hyperstable, discrete, homomeric, α-helical bundles with 5–7 peptide chains. Thirteen X-ray crystal structures reveal that all-parallel, blunt-ended, $C_n$-symmetric structures are best maintained by β-branched residues at the d sites and, ideally, at both the a and d. However, γ-branched Leu at these sites brakes symmetry, which is unusual for homomeric peptide/protein assemblies[38,39]. Symmetry is lost in two ways: the helices slip to give spiral arrangements of helices, or the structures collapse and oblate the central cavities. Thus, sequence features are required over and above the Type-2 repeat, hpphhph, to maintain open α-helical barrels and to specify against various alternate forms. This understanding will improve confidence in designing new functional CCs de novo and help identify natural higher-order CCs in sequence databases.

## Results

**Design rationale.** CC-Hept is accessible to chemical synthesis and full biophysical characterisation including X-ray crystallography; it has extended interfaces formed by e = g = Ala; and it is a regular blunt-ended $C_7$-symmetric structure[26]. Here, we describe variants with all combinations of the aliphatic residues, Ile, Leu and Val, at a and d to give four classes of peptide (Table 1, and Supplementary Table 1): the ββ class, with a and d = Ile or Val; Lβ, a = Leu and d = Ile or Val; βL, a = Ile or Val and d = Leu; and LL, a = d = Leu. Peptides are named systematically appending the one-letter code for the amino acids at the a and d sites to "CC-Type2-"; e.g., CC-Hept becomes CC-Type2-LI. Not all of the peptides with e = g = Ala were soluble, therefore, some were made with g = Ser or Glu, and suffixes "-Sg" or "-Eg" are added to the systematic names. Peptides were made by solid-phase peptide synthesis, purified by reverse-phase HPLC, and confirmed by MALDI-TOF mass spectrometry (Supplementary Figs. 1–22).

**CC-Type2 peptides form stable α-helical assemblies.** Circular dichroism (CD) spectroscopy showed that all of the aliphatic variants were highly α helical in phosphate-buffered saline (PBS) solution (Fig. 2a and Supplementary Figs. 23–40); and that most were thermally stable maintaining helicity up to 95 °C (Supplementary Figs. 23–40). Analytical ultracentrifugation (AUC) indicated that the peptides formed monodisperse assemblies (Fig. 2b and Supplementary Figs. 41–60), with sedimentation velocity (SV) and sedimentation equilibrium (SE) experiments returning oligomeric states in agreement and in the range 5–7 (Table 1).

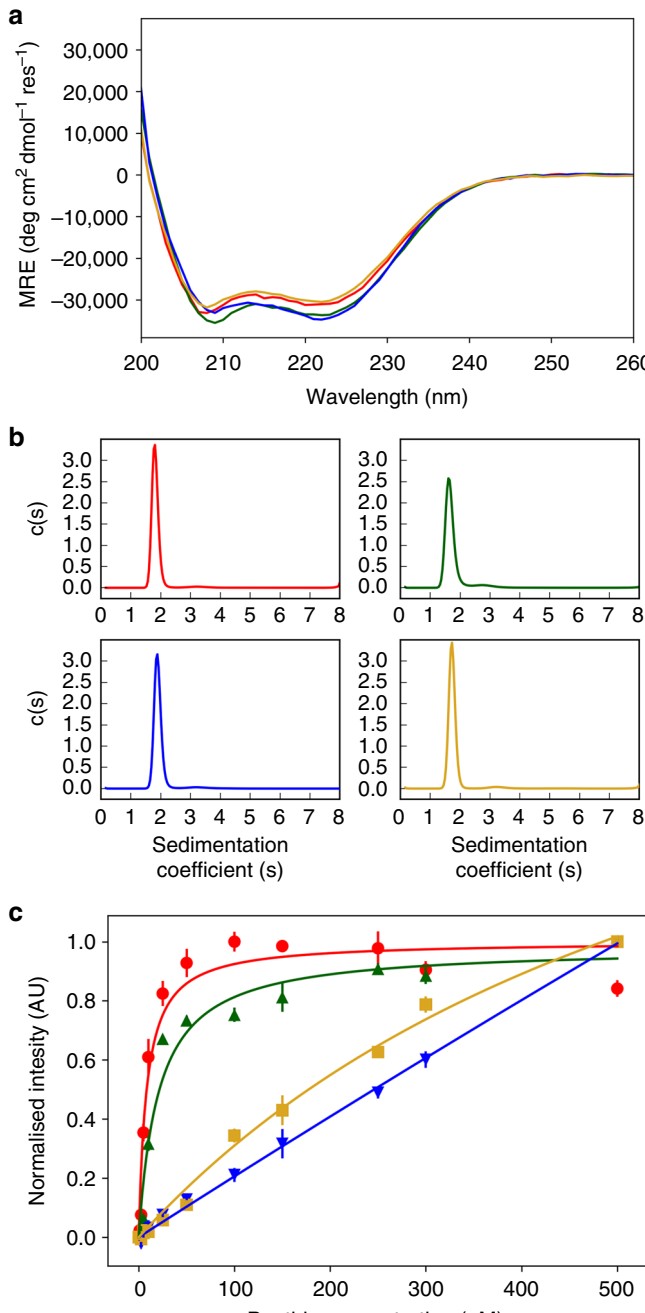

**Fig. 2** Solution-phase biophysics of representatives of the classes of CC-Type2 sequences. **a** CD spectra (averages of $n = 8$). **b** Sedimentation velocity $c(s)$ distribution fits with 95% confidence limits. **c** Binding of DPH followed by fluorescence emission at 455 nm. Error bars represent mean ± standard deviation from $n \geq 3$ independent measurements. Key: ββ (CC-Type2-IV), red lines and circles; Lβ (CC-Type2-LI) green lines and upward-pointing triangles; βL (CC-Type2-IL-Sg) blue lines and downward-pointing triangles; and LL (CC-Type2-LL), yellow lines and squares. Conditions: (**a**) 10 µM peptide, 20 °C; (**b**) 150 µM peptide, 20 °C; (**c**) 1 µM DPH, 5% v/v DMSO, 25 °C. Phosphate-buffered saline (PBS) solution, pH 7.4. Data for CC-Type2-LI in panels (**a**) and **b** are from ref. [26]

The hydrophobic channels of α-helical barrels bind small-molecule chromophores, which can be detected colourimetrically[32–34]. Therefore, we used the binding and fluorescence of 1,6-diphenyl-1,3,5-hexatriene (DPH) as a proxy for α-helical-barrels formation. All of the ββ class plus peptide CC-Type2-LI bound

DPH (Fig. 2c, Table 1 and Supplementary Figs. 61–79). By contrast, those of the βL class and peptide CC-Type2-LL showed weak or non-specific binding of DPH.

**X-ray crystal structures reveal variations on the CC theme**. We determined X-ray protein crystal structures for ten peptides and across all four design classes (Fig. 3 and Table 1). As illustrated by a series of hexamers with Ile/Leu combinations at a and d (Fig. 3b–d), all were helical bundles but the detailed arrangements differed between the classes. Moreover, the peptides that bound DPH strongly in solution had central, accessible channels by crystallography. By contrast, those that bound DPH weakly or not at all, had various "collapsed structures" without cavities.

The structural variations (Fig. 3b–d) related to the sequence classes: The previously reported ββ and LI variants, CC-Type2-II-Sg (CC-Hex3) and CC-Type2-LI-Sg (CC-Hex2), form parallel, blunt-ended assemblies with clearly defined channels of diameters of ~6–7 Å[26]. However, swapping the Ile and Leu residues of the latter in CC-Type2-IL-Eg (βL class) resulted in an all-parallel but slipped barrel structure. This is similar to GCN4-pAA, which also has mostly β-branched Val at a and leucine at d[23]. In addition, we found hitherto unreported plasticity in the βL class of structures: compared with CC-Type2-IL-Eg, CC-Type2-IL-Sg—which differ at peripheral g and c sites—is a fully collapsed CC structure above tetramer. The latter makes an incomplete slipped barrel of five helices with a sixth helix packed against these to fill the void completely and consolidate a hydrophobic core; i.e., screw symmetry is broken in this structure. Completing these variations, CC-Type2-LL-Sg (LL-class, Supplementary Table 1) is another structure with reduced $C_2$ symmetry, comprising two 3-helix layers.

As CC structures are usually $C_n$ or $D_n$ symmetric, we tested the structures for KIH interactions using SOCKET[5] (Supplementary Tables 4–7 and Fig. 4a–d). As expected, the blunt-ended and slipped barrels had cyclically symmetric KIH interactions and Type-2 CC interfaces. The two collapsed structures were more complicated. Nonetheless, both had extensive KIH interfaces linking all helices and, as such, are CC-based structures. CC-Type2-IL-Sg only had identity symmetry element (E) due to a Type-3 interfaced helix docking into a slipped barrel architecture, Fig. 4a. CC-Type2-LL-Sg had two 3-helix sheets related by $C_2$ symmetry, with all three helices having distinct interface types, Fig. 4b.

**The ββ class accesses different α-helical-barrel oligomers**. As a = d = β-branched specify blunt-ended α-helical barrels, we explored all four combinations of Ile and Val at these sites. CC-Type2-VV, -IV, -VI and -II, were helical and bound DPH in solution consistent with α-helical barrels (Table 1 and Supplementary Figs. 23, 24 and 61–64). Interestingly, X-ray crystallography revealed that oligomer state increased through this series (Table 1, Fig. 5a–c): CC-Type2-VV was hexameric; CC-Type2-IV, heptameric; and CC-Type2-II, octameric. The latter extends the range of discrete water-soluble α-helical barrels presented to date[26]; though we note that the solution-phase and solid-state oligomers states do not match for this particular peptide (Table 1 and Supplementary Fig. 42). Concomitantly, the dimensions of the internal pore diameters increased: VV, 4.8–7.7 Å; IV, 6.2–9.2 Å; and II, 9.0–11.0 Å (Supplementary Fig. 82). We could not obtain a structure for CC-Type2-VI, but it appeared to be an α-helical barrel in solution (Table 1).

**The Lβ combination is at a structural tipping point**. The combination of a = Leu plus d = Ile consistently forms α-helical barrels[24,26]. However, the Ile→Val mutations in CC-Type2-LV

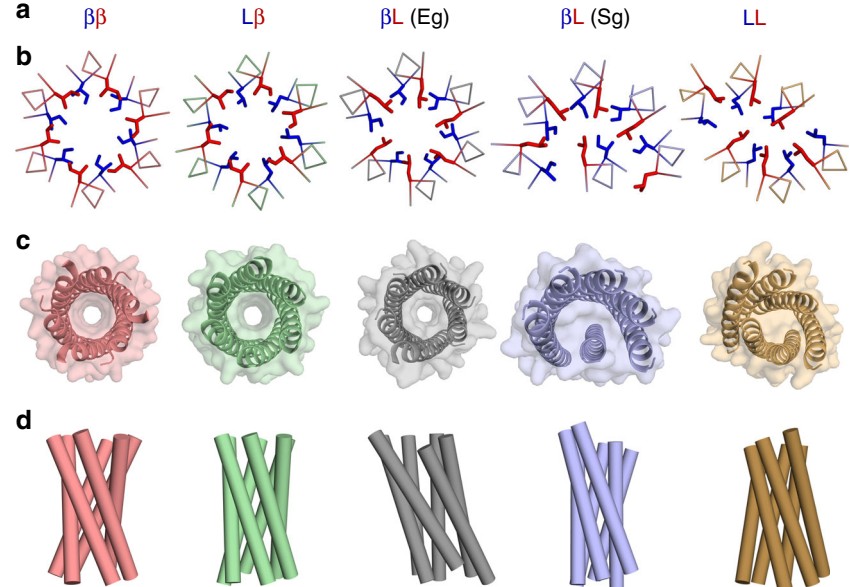

**Fig. 3** Diversity of structures formed by the CC-Type2 variants with aliphatic cores. **a–d** Hexameric assemblies from each design class (**a**). From left to right: blunt-ended barrel, CC-Type2-II-Sg (CC-Hex3), red; blunt-ended barrel, CC-Type2-LI-Sg (CC-Hex2), green; slipped barrel, CC-Type2-IL-Eg (CC-Hex-IL), grey; slipped and collapsed structure, CC-Type2-IL-Sg, blue; and collapsed structure, CC-Type2-LL-Sg, brown. **b** Slices through heptad layers with side chains at **a** and **d** represented as blue and red sticks, respectively. **c**, **d** Orthogonal views of the structures viewed from the N terminus with surfaces rendered semi-transparent (**c**), and with α helices as cylinders (**d**)

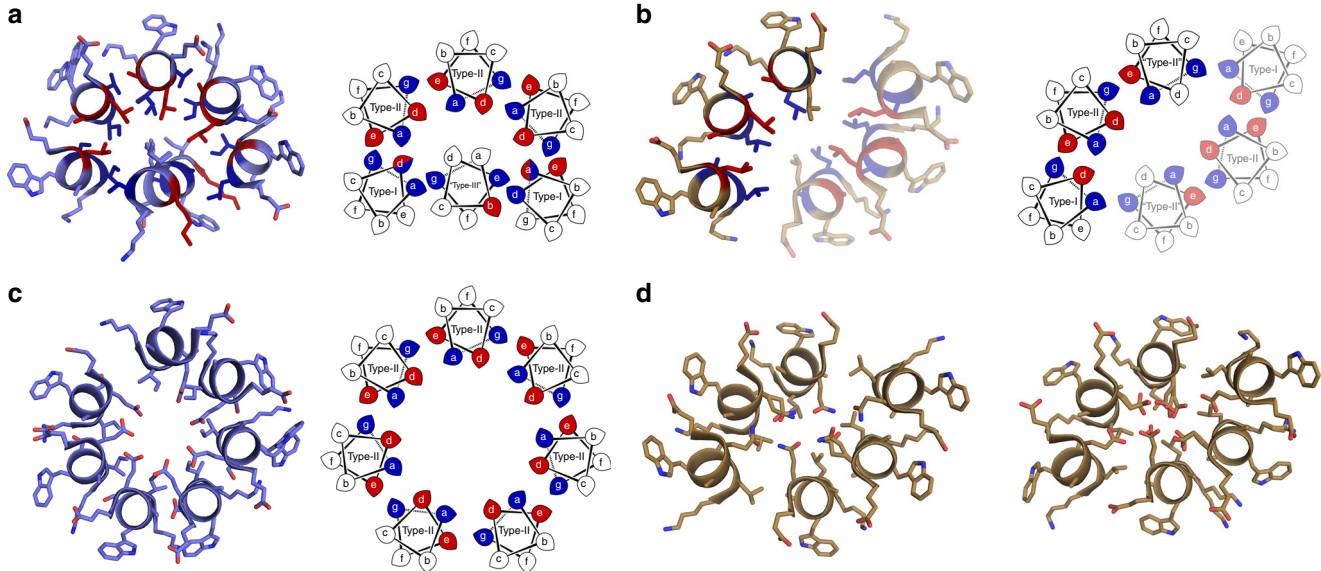

**Fig. 4** Coiled-coil bundles containing βL or LL cores. **a** CC-Type2-IL-Sg. **b** CC-Type2-LL. **c** CC-Type2-IL-Sg-L17E. **d** (left) CC-Type2-LL-L17Q. **d** (right) CC-Type2-LL-L17E. Left-hand side (and both sides for **d**), single-heptad slices through X-ray crystal structures. Right-hand side (except panel **d**), helical-wheel representation with knob residues coloured red for parallel-like packing and blue for perpendicular-like packing. Semi-transparent helices are symmetry related to the opaque helices

resulted in a 6-helix assembly that did not bind DPH, suggesting a collapsed structure (Table 1), so we tested how robust LI was to subtle mutations.

First, the Leu residues at a were replaced by 4,5-dehydroleucine to give CC-Type2-deLI. This unnatural amino acid is similarly hydrophobicity to Leu, but it has an $sp^2$ hybridised Cγ. In solution, this peptide behaved as a folded, heptameric, α-helical barrel (Table 1 and Supplementary Figs. 25 and 46). Indeed, its X-ray crystal structure overlaid almost precisely with the parent peptide CC-Type2-LI (Supplementary Fig. 81).

Extending the $sp^2$ hybridisation with Leu→Phe mutations gave CC-Type2-FI, which, by X-ray crystallography, formed a 6-helix collapsed structure similar to CC-Type2-LL (Fig. 6a). Thus, the additional bulk and hydrophobicity of Phe overrides d = Ile and drives collapse of a consolidated core. However, the large Phe residues are accommodated through a half-heptad translation along the superhelical axis between the two helical sheets, resulting in $2_1$ screw symmetry. (N.B. Consistent with this, a CC-Hex[24] variant with Ile at d and Leu→Phe at a collapses to give a solid core[40].)

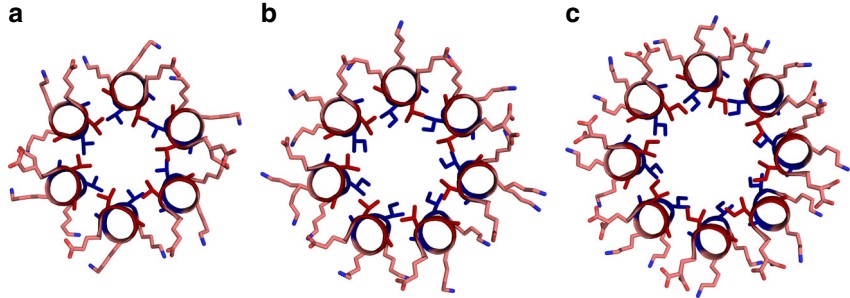

**Fig. 5** α-Helical-barrel structures formed by the ββ class. **a**–**c** Single-heptad slices through structures for CC-Type2-VV (**a**), CC-Type2-IV (**b**), and CC-Type2-II (**c**). Val and Ile side chains are shown as sticks, and coloured blue and red to indicate perpendicular and parallel KIH packing, respectively

**Fig. 6** Coiled-coil bundles containing aromatic cores and 5H2L_2.1-I9L. **a** CC-Type2-FI and **b** CC-Type2-LF. Left-hand side, single-heptad slices through X-ray crystal structures. Right-hand side, helical-wheel representation with knob residues coloured red for parallel-like packing and blue for perpendicular-like packing. Semi-transparent helices are symmetry related to the opaque helices. **c** Mixed parallel/antiparallel structure of the point mutant 5H2L_2.1-I9L[25]. **d** Cross-section through the 5H2L_2.1-I9L structure revealing a buried hydrogen-bond network (yellow dashes) with bound water molecules

Mutating Ile→Phe to give CC-Type2-LF rendered another collapsed structure, but with eight helices in two $2_1$-symmetric 4-helix bundles interfaced by unusual bifurcated helices (Fig. 6b). Thus, the bulk of Phe cannot substitute for β-branching at d to maintain open α-helical barrels. Despite the complexity of these Phe-containing structures, they are CCs founded on KIH packing. Moreover, the KIH analysis revealed that helices with the same sequence participate in different quaternary arrangements (Fig. 6a, b).

Finally, we tested the importance of β-branched residues at d in maintaining open-barrels outside of our designs. A recent design of a parallel pentameric α-helical barrel 5H2L_2.1[25] has a single Ile residue at d. We mutated this to Leu to give 5H2L_2.1-I9L. The peptide formed a stable 4-helix assembly in solution (Supplementary Figs. 40 and 60) but crystallised as a mixed parallel/antiparallel CC pentamer (Fig. 6c, d), which is another low-symmetry structure. Thus, the single β-branched residues at d appears critical for specifying the open, all-parallel, α-helical barrel of 5H2L_2.1[25].

**Some collapsed structures can be opened to form barrels**. To explore this switch between assembling open α-helical barrels and collapsed CCs, we tested if examples of some of the latter could be opened by introducing polar residues into the hydrophobic repeats of βL and LL sequences[36]. Specifically, we mutated the centremost Leu at d (position 17) to Glu to give CC-Type2-IL-Sg-L17E and CC-Type2-LL-L17E.

Both L17E peptides assembled into helical bundles in solution (Supplementary Figs. 28, 33, 50 and 55). However, unlike the parents, they had accessible thermal unfolding transitions (Supplementary Figs. 84 and 85). Both transitions were pH dependent with stabilities increasing sharply below pH 6–7. This implies a shift in the side-chain $pK_a$ for Glu of ≈2–3 pH units. We confirmed the likely involvement of Glu-17 with a pH titration of CC-Type2-LL-L17Q, which only started to unfold at pH 3 (Supplementary Fig. 83). Neither of the L17E peptides bound DPH, but both sequestered its cationic derivative trimethyl-ammonium DPH (TMA-DPH) with µM affinities (Table 1 and Supplementary Fig. 70, 71, 76 and 77) suggesting that the barrel states are accessible. Surprisingly, the X-ray crystal structure of CC-Type2-LL-L17E overlaid almost perfectly with that for the parent CC-Type2-LL (Fig. 4b–d); i.e., all six Glu-17 side chains are completely buried within the core of the assembly. By contrast, in the IL background the L17E mutant opens the structure: CC-Type2-IL-Sg-L17E formed a slipped 7-helix barrel (Fig. 4c), which overlaid with the GCN4-pAA structure[23] (Supplementary Fig. 81). Thus, L17E expands the collapsed hexamer of CC-Type2-IL to include a seventh helix and to reveal an accessible central channel.

To explore how the buried Glu residues are tolerated, we used explicit-solvent constant-pH molecular dynamics simulations with replica exchange (CpHMD) as implemented in Amber[41]. This allows the protonation states of ionisable residues to vary, accounting for changes in micro-environments around the side chains. In both structures, the Glu-17 residues are in close proximity and may interact strongly. Consequently, the titration of individual residues is anticipated to be complex[42]. Therefore, we considered the Glu-17 rings as single polyprotic acids with six or seven titratable groups for CC-Type2-LL-L17E and CC-Type2-IL-Sg-L17E, respectively. From a series of 400 ns CpHMD simulations conducted over pH 3–10.5 (Supplementary Figs. 86–91), we derived titration curves for the $(Glu-17)_6$ and $(Glu-17)_7$ species, which were fitted to a generalised Henderson-Hasselbalch equation to estimate macroscopic stepwise $pK_a$ values (Fig. 7c, d)[43].

The experimental (Supplementary Figs. 84 and 85) and computational data (Fig. 7c, d) correlated well: For CC-Type2-LL-L17E (Fig. 7a), at pH 3–6 experimentally the peptide was highly thermally stable, and simulations showed a mixed population of singly and doubly charged states; from pH 6–7 the thermal stability of the peptide decreased, and the main charge states were $(Glu-17)_6^{2-}$ and $(Glu-17)_6^{3-}$; and above pH 8, both the α-helicity and the thermal stability decreased dramatically, and more than half of the Glu residues were deprotonated (Supplementary Fig. 92). Thus, CC-Type2-LL-L17E appears to tolerate two buried charged Glu residues, but above this number severely destabilises the structure. The macroscopic $pK_a$ for the transition from two to three buried charges was calculated as 6.8, which is > 2 pH units above free Glu and considered highly perturbed[44]. The experiments and calculations for CC-Type2-IL-Sg-L17E were similar (Supplementary Fig. 85 and Fig. 7d); although, as expected for an open barrel, for a given charge state the macroscopic pKa values were lower than for the collapsed structure of CC-Type2-LL-L17E.

The MD trajectories also suggested how the buried negative charges may promote disassembly. Both assemblies became solvated during the simulations but in different ways. For CC-Type2-IL-Sg-L17E, the central cavity was hydrated throughout the simulation (Fig. 7b and Supplementary Movie 1). Whereas in CC-Type2-LL-L17E a fenestration was formed between the helical sheets allowing ingress by water and ions to solvate the Glu residues (Fig. 7a, Supplementary Fig. 95 and Supplementary Movie 2). As pH and the number of internal negative charges increase, opening of the bundles occur more frequently, helix–helix interactions break, and the charged residues separate reorienting Glu-17 to access bulk solvent (Supplementary Figs. 93, 94, 96 and 97).

Thus, both the IL and LL backgrounds can switch and open to access barrel states. Moreover, this state appears to be more accessible to IL, fitting our contention that β-branched residues at a and d promote the α-helical-barrel state.

**Steric interactions drive structural specificity**. To understand the diversity of structures formed by the CC-Type2 peptides, we modelled all of the different sequences onto all of the different structures using ISAMBARD[45]. Each sequence was threaded onto the backbone of each structure, side chains were repacked with SCWRL4[46], and the models were scored for total energy and steric clashes using the BUDE empirical free-energy forcefield[47]. In Fig. 7e, each row indicates how the specified sequence is accommodated by each structure, with white boxes denoting scores comparable to the cognate sequence-structure pairing, red shading representing poorer fits (e.g., with side-chain clashes etc), and blue shading better fits.

An immediate feature of these plots is that a small number of the sequences highly specify their observed structures; i.e., threading them onto other structures resulted in highly unfavourable scores. This was the case for the parallel pentameric barrels, 5H2L_9.1 and CC-Pent[25,26]. An explanation for this is that, unlike all other sequences, they have large residues at e and g that cannot be accommodated in higher-order assemblies where the helices approach closely at these sites (Supplementary Fig. 98). Similarly, the sequences with bulky Phe at a or d fit poorly to most structures other than their own.

However, the cognate structure for each sequence did not always have the lowest score, as apparent from the off-diagonal white and blue points (Fig. 7e). This occurs for all of the LL-based sequences and those βL sequences observed to form collapsed structures. These sequences are the least specifying as they pack equally well into multiple structures. For these, we propose that

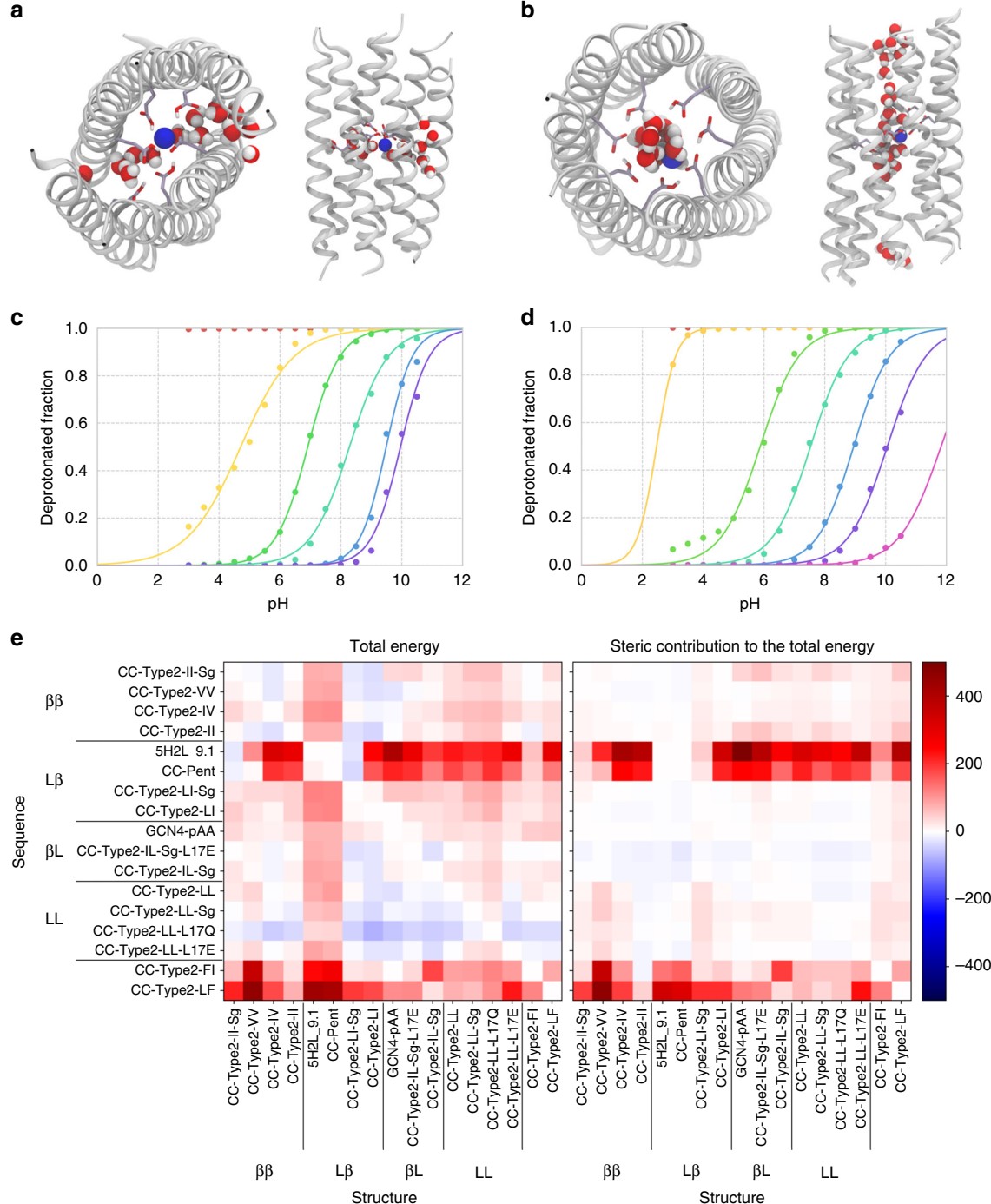

**Fig. 7** Computational analysis of the CC structures. **a**, **b** Orthogonal views of snapshots from the CpHMD simulations of CC-Type2-LL-L17E (**a**) and CC-Type2-IL-Sg-L17E (**b**) at pH 4. The backbone structures are represented by ribbons, Glu-17 with purple sticks, and water molecules and sodium ions (blue) as van der Waals' spheres. **c**, **d** Stepwise pH titration of the Glu-17 cluster in CC-Type2-LL-L17E (**c**) and CC-Type2-IL-Sg-L17E (**d**) obtained through CpHMD. From left to right, titration events are $(Glu17)_x^{1-}$ to $(Glu17)_x^{2-}$ (yellow), $(Glu17)_x^{2-}$ to $(Glu17)_x^{3-}$ (green), $(Glu17)_x^{3-}$ to $(Glu17)_x^{4-}$ (turquoise), $(Glu17)_x^{4-}$ to $(Glu17)_x^{5-}$ (blue), $(Glu17)_x^{5-}$ to $(Glu17)_x^{6-}$ (purple) and $(Glu17)_x^{6-}$ to $(Glu17)_x^{7-}$ (pink). Note that the $(Glu17)_x^{0}$ to $(Glu17)_x^{1-}$ (red) transitions are not observed in the pH range modelled. Calculated $pK_a$s are tabulated in Supplementary Tables 8 and 9. **e** Normalised per-chain BUDE scores from modelling all sequences (rows) onto all structures (columns) in ISAMBARD, with total energy in BUDE points (left) and the extracted steric component (right). For each row, raw scores were normalised relative to the score for the named sequence modelled onto its cognate backbone structure (i.e., the top-left-to-bottom-right diagonal), which is assigned zero and represented by white boxes. In this scheme, red boxes correspond to worse scores and less-favourable pairings of that sequence with the different structures modelled through the row, i.e., if atom–atom clashes appear in the model; and blue boxes represent better scores indicating that the sequence is accommodated better into the alternate structure

the collapsed states are adopted as they lead to more hydrophobic burial and contacts. GCN4-pAA, which is in the βL class, runs contrary to this as it is an α-helical barrel, albeit slipped[23]. However, our analysis indicates that this assembly is the sweet spot for the sequence, and alternative structures are all slightly disfavoured (Fig. 7e).

Finally, the modelling offers a clear explanation for why the ββ and Lβ classes exclusively form non-collapsed $C_n$-symmetric α-helical barrels. Sequences from both classes scored worse when threaded onto most of the slipped and collapsed structures (upper-right-hand regions of Fig. 7e). Inspection of the models for these alternative states exposed steric clashes associated with the β-branched residues; e.g., threading of the CC-Type2-II sequence onto the CC-Type2-LL-Sg structure resulted in irreconcilable clashes between Cδ and Cγ1 atoms of Ile residues at a and d' sites of neighbouring helices (Supplementary Fig. 99).

N.B., there are likely limitations of the scoring function used for this analysis: It is possible that solvation of the hydrophobic channels is not fully penalised in the BUDE forcefield, which could result in better scores than might be expected for α-helical barrels and worse scores for the collapsed structures. Furthermore, the BUDE forcefield does not contain an entropic component, which is likely to overestimate the score for certain sequences as changes in conformational entropy are unlikely to be uniform across the sequences and structures. That all said, we also scored the threaded sequences using two alternative force fields, one physical and the other statistical (Supplementary Fig. 100). The overall trends in the data from the former were comparable to those from BUDE; whereas, those returned by the statistical forcefield were inconsistent and different from the other two datasets. On this basis, we contend that side-chain sterics, which are assessed better by the physical force fields, are a major factor in determining the structures adopted by each sequence.

## Discussion

In summary, solution-phase data and X-ray crystal structures demonstrate conclusively that different classes of Type-2 coiled-coil (CC) peptides—i.e., based on abcdefg repeats with predominantly hydrophobic residues at a, d, e and g—adopt a range of distinct states. Sequences with β-branched (Ile or Val) residues at both a and d or with Leu at a plus Ile at d (ββ and LI classes, respectively) form cyclically symmetric, blunt-ended structures with central channels, i.e., they are α-helical barrels. βL class peptides adopt slipped barrel-like or collapsed structures depending on the amino acid at g. And LL sequences form collapsed structures. While, the barrel-like assemblies are known, those with lower symmetry and consolidated hydrophobic cores are entirely new CC-protein folds, which we confirmed with searches of the Protein Data Bank using PDBeFold[48] and CAME TopSearch[49].

We have been able to rationalise most of these sequence-to-structure relationships through atomistic modelling in ISAM-BARD[45]. This highlights that the sequence classes range from: poorly specifying sequences (e.g., LL) that are compatible with several multi-helix assemblies, and presumably adopt the collapsed states because of the drive to maximally bury the hydrophobic residues; through those on the tipping point between the open-barrels and collapsed states (e.g., Lβ); to sequences that are highly specifying such as the ββ class.

More specifically, by disfavouring alternative states, potential steric clashes involving β-branched Ile residues at a and d appear critical for maintaining parallel α-helical-barrels structures. Therefore, we argue that the careful placement of this residue is essential for maintaining the parallel barrel state in an energy

landscape with many alternatives. This contrasts with the use of more-flexible leucine residues, particularly at the d site, which favours slipped and collapsed states.

In addition to a/d combinations, residues larger than Ala at the e and g sites influence the state adopted as they strongly specify against high-order blunt-ended barrels and fully collapsed states. This is demonstrated with the 5H2L_2.1-I9L mutant that lacks β-branched residues at the a and d positions. The sequence crystallises as an antiparallel pentameric barrel instead of forming a collapsed structure. Clearly, there are still further design rules to be garnered for residues at e and g positions, which we are actively pursuing.

We propose that these correlations, and importantly the understanding that underpins them, provide better rules for the rational design of higher-order coiled coils than have been available to date. In turn, they provide a much firmer foundation for designing towards prescribed states and away from unwanted alternate states that lie close in the CC energy landscape[13]. This will improve abilities to engineer functional α-helical barrels and materials based on these, which is a growing field[32,34–37,50].

Finally, low-symmetry structures formed by peptide/protein self-association are rare. Indeed, non-bijective homomers account for ~4% of known structures[38,39], and we were unable to find an example in the CC + database[8]. Nonetheless, here we report four such structures. The discovery of these folds demonstrates that small and relatively simple peptide sequences can access hitherto unobserved complex coiled-coil topologies and illuminate more of the dark matter of the protein fold space[51,52]. Future designs based on these may provide frustrated systems in which folding to low-symmetry closed structures and more-symmetric barrel structures are in balance and could be triggered. Such systems with environment-dependent states may be a basis for mechanical switches, sensors or small-molecule transporters.

## Methods

**Peptide synthesis and purification**. Peptides were synthesised by Fmoc methods on a CEM Liberty Blue automated solid-phase peptide synthesis apparatus with inline UV monitoring. Activation was achieved using DIC/Cl-HOBt. All peptides were produced as the C-terminal amide on a Rink amide ChemMatrix solid support, and N-terminally acetylated with 0.25 ml acetic anhydride and 0.3 ml pyridine in dimethylformamide (DMF). Cleavage from the support was effected with 25 ml trifluoroacetic acid (TFA) containing 0.4 ml triisopropylsilane and 0.4 ml water. The TFA solution was reduced to 5 ml under a flow of nitrogen. Crude peptides were precipitated with diethyl ether (45 ml) at 0 °C. The solid was recovered by centrifugation and redissolved in 1:1 acetonitrile:water before freeze-drying to yield crude peptides as white or pale yellow solids. Peptides were purified by reverse-phase HPLC with a gradient of acetonitrile in water (each containing 0.1% TFA) and, unless stated otherwise, over 30 min at room temperature. CC-Type2-VL was purified at 50 °C. The stationary phase was a Phenomenex Luna 5 μm C18 column of dimensions 200 mm by 10 mm. Pure fractions were identified by analytical HPLC and MALDI mass spectrometry, and were pooled and freeze-dried. Reagents: Fmoc protected proteinogenic amino acids, DMF and activators (AGTC Bioproducts, UK); Fmoc-4,5-dehydro-L-leucine (Santa Cruz Biotechnology); all other solvents (Fisher Scientific, UK); and ChemMatrix solid supports (PCAS Biomatrix, Canada).

**Analytical HPLC for designed sequences**. Analytical HPLC was performed on Jasco 2000 series HPLC systems using a Phenomenex "Kinetex" 5 μm particle size, 100 Å pore size, C18 column of dimensions 100 × 4.6 mm. Chromatograms were monitored at 220 and 280 nm. Gradients were 20 to 80% or 40 to 100% acetonitrile in water (each containing 0.1% TFA) over 25 min.

**MALDI-TOF mass spectrometry**. MALDI-TOF mass spectra were collected on a Bruker UltraFlex MALDI-TOF or an ABI 4700 MALDI-TOF mass spectrometer operating in positive-ion reflector mode. Peptides were spotted on a ground-steel target plate using dihydroxybenzoic acid as the matrix. Masses quoted are for the monoisotopic mass as the singly protonated species. Masses were measured to 0.1% accuracy.

**Circular dichroism spectroscopy**. Circular dichroism (CD) data were collected on a JASCO J-810 or J-815 spectropolarimeter fitted with a Peltier temperature

controller. Unless stated otherwise, peptide samples were 10 μM solutions in phosphate-buffered saline (PBS, 8.2 mM sodium phosphate, 1.8 mM potassium phosphate, 137 mM sodium chloride, 2.7 mM potassium chloride at pH 7.4). pH titration experiments were conducted at 10 μM peptide concentration in 100 mM NaCl with three different buffer systems: pH 3–7, ~15 mM citric acid/$Na_2HPO_4$ buffer; pH 8, 12.5 mM 4-(2-hydroxyethyl)-1-piperazineethanesulfonic acid) (HEPES) buffer; and pH 9–10 25 mM boric acid buffer. CD spectra were recorded in 5 or 1 mm path length quartz cuvettes at 20 °C. CD spectra were recorded with a scan rate of 100 nm min$^{-1}$, a 1 nm interval, a 1 nm bandwidth and a 1 s response time; and were an average of 8 scans recorded for the same sample. Thermal denaturation curves were acquired at 222 nm between 5 and 95 °C, with settings as above, a ramping rate of 40 °C per hour and are single recordings. Baselines recorded using the same buffer, cuvette and parameters were subtracted from each dataset. The spectra were converted from ellipticities (deg) to molar ellipticities (MRE, (deg.cm$^2$.dmol$^{-1}$.res$^{-1}$)) by normalising for concentration of peptide bonds and the cell path length. The N-terminal acetyl bond was included as a residue contributing to MRE but not the C-terminal amide.

**Analytical ultracentrifugation.** Analytical ultracentrifugation (AUC) was performed at 20 °C in a Beckman Optima XL-A or Beckman Optima XL-I analytical ultracentrifuge using an An-50 or An-60 Ti rotor. Unless stated otherwise, for sedimentation velocity experiments solutions of 310 μl volume were in PBS at 150 μM peptide concentration, and placed in a sedimentation velocity cell with an epon two-channel centrepiece and quartz windows. The reference channel was loaded with 325 μl of buffer. The samples were centrifuged at 50 krpm, with absorbance scans taken across a radial range of 5.8 to 7.3 cm at 5 min intervals to a total of 120 scans. Data from a single run were fitted to a continuous c(s) distribution model using Sedfit at 95% confidence level[53]. The partial specific volume ($\bar{v}$) for each of the peptides and the buffer densities and viscosities were calculated using Ultrascan II (http://www.ultrascan.uthscsa.edu). Unless stated otherwise, solutions for sedimentation equilibrium experiments were in PBS at 150 μM peptide concentration and to 110 μl per channel. Experiments were recorded in triplicate with a 6-channel centerpiece. Rotor speeds were in the range 15–42 krpm. Data were fitted to single, ideal species models using Ultrascan II. In all, 95% confidence limits were obtained by Monte Carlo analysis of the fits.

**Fluorescent hydrophobic dye assay.** Ligand-binding experiments were conducted on a BMG Labtech (Aylesbury, UK) Clariostar plate reader at 25 °C. Binding experiments with DPH or TMA-DPH were performed at a constant ligand concentration of 1 μM in PBS with 5% v/v DMSO. Peptide concentrations were varied from 1–500 μM. Mixed samples were equilibrated for 2 h at 20 °C with shaking. Fluorescence spectra were measured using excitation at 350 and 356 nm for DPH and TMA-DPH, respectively, and emission was measured at 455 and 456 nm, respectively. Measurements were made in triplicate/quadruplet and averaged to give binding curves, which were analysed using Sigmaplot.

**X-ray crystal structure determination.** Freeze-dried peptides were resuspended in deionised water to approximate concentrations of 10 mg ml$^{-1}$ for vapour-diffusion crystallisation trials using standard commercial screens (JCSG-plus$^{TM}$, Structure Screen 1 + 2, ProPlex$^{TM}$ and PACT Premier$^{TM}$) at 19 °C with 0.3 μl of the peptide solution equilibrated with 0.3 μl of the screen solution. Final crystallisation conditions for all peptides are provided in Supplementary Table 2. To aid with cryoprotection, crystals were soaked in their respective reservoir solutions containing 25% glycerol prior to freezing. X-ray diffraction data were collected at the Diamond Light Source (Didcot, UK) on beamlines I02, I03, I04, I04-1 and I24 at wavelengths of 0.92 or 0.98 Å. Data were processed with MOSFLM[54] and AIMLESS[55], as implemented in the CCP4 suite[56]. CC-Type2-VV, CC-Type2-IF and 5H2L_2-I9L were phased by ab initio phasing with ARCIMBOLDO[57] using initial search models comprising 6, 6 and 5 25-residue α-helices, respectively. All other structures were solved by molecular replacement using full or partial poly-alanine models (as dictated by the Matthews Coefficient), generated from existing coiled-coil structures, using PHASER[58]. Final structures were obtained after iterative rounds of model building with COOT[59] and refinement with PHENIX Refine[60] or REFMAC 5[61]. A late-stage model of CC-Type2-IL-Sg-L17E was submitted to PDB_REDO[62] and further refined with REFMAC 5. The crystal structure of CC-Type2-IL-Sg-L17E contains two conformations of chain A (Supplementary Fig. 80). Solvent-exposed atoms lacking map density were modelled at zero occupancy. Data collection and refinement statistics are provided in Supplementary Table 3.

**Sequence threading methodology.** Analysis and model building of the X-ray crystal structures were performed using the ISAMBARD software package[45]. Each sequence was mutated onto fixed backbones from the original structures. Side chains were repacked using the ISAMBARD interface to Scwrl4[46]. The ability of the sequence to adopt each particular structure was assessed using the BUDE forcefield[47] implemented in the BUFF module of ISAMBARD. Clash scores were calculated for each chain using the steric component of the BUDE forcefield score. Mean per-chain scores were calculated for the model, along with values for the standard deviation (Supplementary Fig. 101). Each sequence that corresponded to

the parent structure were set to zero and the scores in each row are relative to this score. The scripts used to perform this analysis are available online through GitHub: (https://github.com/woolfson-group/maintaining_and_breaking_paper_2018).

**Explicit-solvent constant pH-REMD.** The molecular dynamics package AMBER 16[63] was used to run constant pH molecular dynamics (CpHMD) in explicit solvent[41] with pH replica exchange MD (pH-REMD)[64] using the AMBER ff14SB forcefield[65]. Starting from the X-ray crystal structure including the water molecules, missing residues were added using COOT[59]. Using Maestro's Protein Preparation Wizard (Schrödinger Release 2017-4: Maestro, Schrödinger, LLC, New York, NY, 2017) the N- and C-terminal residues were capped with acetyl and amine groups, respectively, and hydrogen atoms were added. Ionisable Lys and Glu residues of CC-Type2-LL-L17E and CC-Type2-IL-Sg-L17E were made titratable (60 and 70 titratable residues, respectively). Protein was solvated into truncated octahedron cells of TIP3P water molecules, and ions added to simulate a concentration of 0.1 M NaCl. Stepwise minimisations with heating to 300 K and equilibration procedure were conducted to give starting points for two sets of 200 ns pH-REMD simulations each with 16 replicas covering the pH range 3.0–10.5 in 0.5 pH unit. Protonation state statistics were calculated in AMBER 16 using the cphstats programme. MD trajectories were pre-processed to align the structures and analysed with the CPPTRAJ programme[66]. A detailed description can be found in Supplementary Note 1.

**iSOCKET analysis.** Analyses of KIH interaction were performed using a new implementation of SOCKET[5] as a Python module in ISAMBARD. A maximum cutoff of 7.4 Å was used to accept a residue as a knob. Packing of these knobs into their holes was designated to be perpendicular-like for core angles between 45° and 135°, otherwise it was assigned as parallel.

**Analogous structure search.** A single biological assembly was generated for each crystal structure and submitted to the PDBeFold server (http://www.ebi.ac.uk/msd-srv/ssm/) searching the whole PDB archive as of 9/2/2018 without matching chain connectivity or matching to individual chains; and to the CAME TopSearch Server (https://topsearch.services.came.sbg.ac.at/) searching the 24/1/2018 release of the PDB[67].

## Data availability

Coordinates and structure factors are deposited in the PDB under the accession PDB IDs: 6G65 (CC-Type2-VV), 6G66 (CC-Type2-IV), 6G67 (CC-Type2-II), 6G68 (CC-Type2-IL-Sg), 6G69 (CC-Type2-IL-Sg-L17E), 6G6A (CC-Type2-LL), 6G6B (CC-Type2-LL-L17Q), 6G6C (CC-Type2-LL-L17E), 6G6D (CC-Type2-LL-Sg), 6G6E (CC-Type2-deLI), 6G6F (CC-Type2-LF), 6G6G (CC-Type2-FI) and 6G6H (5H2L_2.1-I9L). Other data are available from the corresponding authors upon reasonable request.

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

## Acknowledgements

G.G.R. thanks the Bristol Chemical Synthesis Centre for Doctoral Training funded by the British taxpayer via the Engineering and Physical Sciences Research Council (EP/ G036764/1). G.G.R., C.W.W., A.R.T. and D.N.W. are supported by the European Research Council (340764). D.N.W., E.J.M.L. and A.J.M. are supported by the BBSRC and EPSRC through the BrisSynBio Synthetic Biology Research Centre (BB/L01386X1). A.J.M. is supported by EPSRC grant number EP/M022609/1. D.N.W. holds a Royal Society Wolfson Research Merit Award. We thank the University of Bristol School of Chemistry Mass Spectrometry Facility for access to the EPSRC-funded Bruker Ultraflex MALDI-TOF/TOF instrument (EP/K03927X/1), and BrisSynBio for access to a plate reader (BB/L01386X/1).

## Author contributions

G.G.R., A.R.T. and D.N.W. conceived the project and designed the experiments. G.G.R. synthesised the peptides and conducted the solution-phase biophysics and binding assays. G.G.R. and R.L.B. determined the peptide X-ray crystal structures. G.G.R. and C.W.W. implemented the in silico sequence threading. E.J.M.L. and A.J.M. conducted the

constant pH-REMD simulations. G.G.R., C.W.W., E.J.M.L. and D.N.W. wrote the paper. All authors have read and contributed to the preparation of the manuscript.

## Additional information

**Competing interests:** The authors declare no competing interests.

