## [Peer Review File · Nature Communications]

Reviewers' comments:

Reviewer #1 (Remarks to the Author):

Rhys et al., have carefully performed a significant amount of work to decipher the rules governing coiled-coil barrels. This work is a natural extension of their foundational helix barrel work. In this work, the authors carefully and systematically examine the effect of amino acids at a, d, e & g positions in the abcdefg coiled coil motif for the classic knob-into-hole packing of 2 alpha helices. The work is impressive and complete from biophysical characterizations to crystal structures to computational analysis. I only have a few major concerns.

First, I disagree with the conclusions from the computational studies. While I am a firm believer in computational analyses, they have a limit that becomes clearly evident when exposed to experimental data. As mentioned, the computational energies do not consider a number of factors, and furthermore are statistically based. So, they are inherently biased by their training set, and furthermore are an average reflection of the training set structures. In Figure 5, the lowest energy should be the diagonal. Any energies that are lower off-diagonal are more likely artefacts of the energy function. That the authors found crystal structures as well as the biophysical analyses strongly argues against certain sequences taking on alternate states. Perhaps under different conditions, but the energy function doesn't consider such cases. Figure 5 reveals problems with the energy functions, but that is more the responsibility of the group that developed the computational functions. In truth, this paper provides an outstanding data set to improve these scoring functions.

Secondly, in the first and second paragraph of the discussion of the effect of the g position should be located to its own paragraph. Looking back over the manuscript, the results mostly focus on the a and d positions. Clarification is needed that relates to where in the results that the g position has importance as well as a discussion of the effects of different residues to this position.

Perhaps a solution to the previous lies in the next point of discussion. As the authors well know, the abcdefg nomenclature refers to the packing between 2 coiled coils, where the a & d positions pack. But the discussion of packing between more of a helix with 2 others becomes more complicated. As an example, the hpphph or hpphph; hpphph, etc notation is somewhat confusing, since the packing register shifts along the abcdefg repeat. In fact, it is the repeat that shifts register between the packing of helix 1 with helix 2 and the packing of helix 1 with helix 3. So, the g residue is actually an a or d position packing into an alternate helix. An alternative that could be used would be to use the ridge concept developed by Chothia, Levitt & Richardson (1981) that was also nicely illustrated in Figure 2 of the corresponding author's previous paper Walshaw & Woolfson (2003). The classic knob-into-holes packing alternates between an i+4 ridge (a & d positions) and a i+3 ridge (d & a+1 positions), where a+1 is the a in the next heptamer repeat. These ridges are indirectly pointed out by the spacing between residues in the 2nd paragraph of the paper. So, in figure 1, the packing of Type 2 helical barrels could be described as the i+4 ridge between residues e-1 and a (although the figure depicts a and e in the same heptad repeat, which would not be classical knob-into-holes packing), whereas the d and g packing are the i+3 ridge packing. Explaining the packing in this respect might help clarify the faces of the helix that are packing. Alternatively, the relative positions of the heptad repeat registers could be used. In the Type 2 packing, the registers are +4 shifted from each other.

On a more minor note, I could not find the reason for listing the heptad repeat starting with the g position in Table 1 as well as what amino acids filled position f.

Lastly, Crick's original 1953 knobs-into-holes paper only referred to a knob repeating every 7 residues. It would be good to reference the first paper denoting the "abcdefg" for the heptamer repeat as well as the identification of the a & d residues as the knobs in the packing.

In summary, this work contributes fundamentally to understanding protein structure prediction. The analysis was performed systematically. Moreover, the wealth of supplementary data was presented in a very organized that is easily digested.

Reviewer #2 (Remarks to the Author):

Rhys et al describe the design and characterization of several families of novel coiled coil sequences and structures. The new macromolecular assemblies are characterized in detail, both in solution and as crystal structures. The structures show that slight changes in sequence can yield completely different oligomeric states. Moreover, some of the assemblies form barrels with internal pores, while others form collapsed structures with little or no interior cavity. Interestingly, some of the structures are symmetric (as would be expected from homo-oligomers), while others are (surprisingly) non-symmetric. The results are compelling and the take-home messages are interesting.

The paper is well-written and is a pleasure to read. An enormous amount of experimental work was done, and it is described clearly and concisely. Importantly, the manuscript does not merely list a bunch of sequences and structures. Instead, the authors do an excellent job explaining how/why subtle changes in sequence - sometimes involving just a few atoms - give rise to different packing and ultimately to dramatically different 3D structures.

This work demonstrates how far the field of de novo protein design has advanced in recent years.

Reviewer #3 (Remarks to the Author):

The paper is a follow-up to an earlier work by the same authors on the design of α -helical barrels. Such structures have the KIH packing and are therefore classified as coiledcoils, albeit very unusual ones. The authors depart from their previous design CC-Hept (which gets a new name here). This structure is based on a 30-mer peptide with the heptad sequence (gabcdef) A?KE?A* , where "?" stands for the residues in a and d positions. The authors vary the latter systematically and analyze the resulting structures. This gives 9 possibilities, of which 3 could not be crystallized. Of the remaining 6 structures (some of which have been previously published) form four symmetrical α -helical barrels. The remaining two form asymmetric "collapsed" structures, i.e. without a pronounced channel in the middle. Some additional variations are also analyzed.

Based on this data, the authors suggest that symmetric α -helical barrels of this type are stabilized by b-branched residues in a and d positions. This prevents them from collapsing and forming the (logically expected) hydrophobic core. Some computational modeling is done, and seems to support the conclusions.

The work contains interesting new data and conclusions that advance the field (design of artificial α -helical barrels). The various asymmetric oligomers described are in fact unusual and rare and thus interesting. My most important remarks are as follows.

I found the paper quite difficult to read. This is partly due to the complexity of the phenomena studied, many of which can not be fully explained. However, a condensed style of writing as well as suboptimal preparation of display elements further aggravate the problem.

In particular, I believe that the figures 3 and 4 need to be carefully revised, as indeed crucial for the understanding. The reader is lost in all these multiple panels, some of which closely overlap in the message. The panels must also be rearranged in a more logical order (for example, a

multipanel figure with symmetric structures, another one with collapsed structures, and another for special cases such as aromatics). At present it is extremely difficult to correlate the figures and Table 1.

The central Table 1 can be improved by adding the column describing the assembly type (symmetric/asymmetric, barrel/collapsed etc) and a column with the PDB code. The old and new names of some assemblies should also be included.

Can the authors provide some quantification of how dense the cores of the individual collapsed structures are? Are these as dense as in globular proteins? Some panels (e.g. Fig. 3b, right; Fig. 4b) suggest the opposite, but it is impossible to tell from the figures alone.

The first 1.5 pages of the introduction (including most of Figure 1) can be sacrificed. It makes little sense to provide this supercondensed overview of all possible coiledcoil structures. The introduction should rather focus on the construction of α -helical barrels. It should be explained which sequence pattern and why was used to come to the structure CC-Hex, the starting point of the current project.

Here I suggest adding an appropriate 3D / depth-rendered view showing the involvement of sidechains of all six (gabcde) positions in the barrel stabilization. The flat, schematic drawings like the panel 3b (and many similar) hardly help.

Shortening the introduction should provide room for more detail throughout the manuscript, which would save the reader the trouble of looking up every single reference before the logic can be understood!

The support obtained by computation is interesting. However, since the result (=experimental structures) is known, the risk of biased conclusions is large. To me, much of the argumentation provided falls short of being strictly causative. Some more mechanistic and detailed explanations, if available, could help. (I liked the 'Computational protein threading' (last section), but here the authors themselves warn on the limitations of the scoring used).

It is also not very clear how far the conclusions can be applied outside of the very specific (A?KE?A*) pattern. The given single example beyond this pattern, that of the modified 5H2L structure, which also does not seem to completely collapse, is not too convincing.

Response to Reviewers' Comments

Maintaining and breaking symmetry in homomeric coiled-coil assemblies

Guto G. Rhys, Christopher W. Wood, Eric J.M. Lang, Adrian J. Mulholland, R. Leo Brady, Andrew R. Thomson, and Derek N. Woolfson

Reviewer 1

Rhys et al., have carefully performed a significant amount of work to decipher the rules governing coiled-coil barrels. This work is a natural extension of their foundational helix barrel work. In this work, the authors carefully and systematically examine the effect of amino acids at a, d, e & g positions in the abcdefg coiled coil motif for the classic knob-into-hole packing of 2 alpha helices. The work is impressive and complete from biophysical characterizations to crystal structures to computational analysis. I only have a few major concerns.

First, I disagree with the conclusions from the computational studies. While I am a firm believer in computational analyses, they have a limit that becomes clearly evident when exposed to experimental data. As mentioned, the computational energies do not consider a number of factors, and furthermore are statistically based. So, they are inherently biased by their training set, and furthermore are an average reflection of the training set structures. In Figure 5, the lowest energy should be the diagonal. Any energies that are lower off-diagonal are more likely artefacts of the energy function. That the authors found crystal structures as well as the biophysical analyses strongly argues against certain sequences taking on alternate states. Perhaps under different conditions, but the energy function doesn't consider such cases. Figure 5 reveals problems with the energy functions, but that is more the responsibility of the group that developed the computational functions. In truth, this paper provides an outstanding data set to improve these scoring functions.

Response 1 - We agree with Reviewer 1 that it is important to evaluate the limitations of computational studies. For the sequence-threading section, the side-chains were packed onto fixed-backbones of the crystal structures using SCWRL4 and each model was scored using the BUDE force field. Firstly, SCWRL4 uses backbone-dependent rotamer libraries to pack each side chain, which is statistically based. That said, candidate rotamers are evaluated for steric clashes using a fast collision detection algorithm followed by a physics-based energy function that emulates a Lennard-Jones potential. Secondly, the BUDE force field is an empirical force field containing a physical energy function to evaluate van der Waals' contacts. The argument for specificity in this paper is based on steric interactions and as such it is free from dataset statistical biases.

We acknowledge that sequences that have better scores off-diagonal demonstrate the limitations of force fields more generally. This is further complicated for self-associating species, which have greater degrees of freedom, compared with monomeric species, for which the environment can affect the interactions and consequently the structures formed. In the manuscript, we offer suggestions why better scores are observed off-diagonal. To demonstrate that this is a limitation of force fields more generally we have rescored all the models using two other force fields, and we have included these data in the revised supplementary information. We have also added the following sentences to page 12 of the manuscript on this.

“That all said, we also scored the threaded sequences using two alternative force fields, one physical and the other statistical (Supplementary Fig. S8). The overall trends in the data from the former were comparable to those from

BUDE; whereas, those returned by the statistical force field were inconsistent and different from the other two datasets. On this basis, we contend that side-chain sterics, which are assessed better by the physical force fields, are a major factor in determining the structures adopted by each sequence.”

Secondly, in the first and second paragraph of the discussion of the effect of the g position should be located to its own paragraph. Looking back over the manuscript, the results mostly focus on the a and d positions. Clarification is needed that relates to where in the results that the g position has importance as well as a discussion of the effects of different residues to this position.

Response 2 - For the most part, our study focuses on the effects of residues at **a** and **d** positions of Type-2-interfaced coiled coils. However, residues **e** and **g** positions do and will alter specificity. This is evident from the sequence-threading analysis for sequences that form pentameric structures, which have large residues **e** and **g** and score poorly when threaded onto most other structures. We are actively researching this, but feel it is beyond the scope of the present paper. To note this, in the revised paper we have reorganised and elaborated this in the discussion on page 13:

“In addition to a/d combinations, residues larger than Ala at the e and g sites influence the state adopted as they strongly specify against high-order blunt-ended barrels and fully collapsed states. This is demonstrated with the 5H2L_2.1-I9L mutant that lacks β -branched residues at the a and d positions. The sequence crystallises as an antiparallel pentameric barrel instead of forming a collapsed structure. Clearly, there are still further design rules to be garnered for residues at e and g positions, which we are actively pursuing.”

Perhaps a solution to the previous lies in the next point of discussion. As the authors well know, the abcdefg nomenclature refers to the packing between 2 coiled coils, where the a & d positions pack. But the discussion of packing between more of a helix with 2 others becomes more complicated. As an example, the hpphph or hpphhpp; hpqhph, etc notation is somewhat confusing, since the packing register shifts along the abcdefg repeat. In fact, it is the repeat that shifts register between the packing of helix 1 with helix 2 and the packing of helix 1 with helix 3. So, the g residue is actually an a or d position packing into an alternate helix. An alternative that could be used would be to use the ridge concept developed by Chothia, Levitt & Richardson (1981) that was also nicely illustrated in Figure 2 of the corresponding author's previous paper Walshaw & Woolfson (2003). The classic knob-into-holes packing alternates between an i+4 ridge (a & d positions) and a i+3 ridge (d & a+1 positions), where a+1 is the a in the next heptamer repeat. These ridges are indirectly pointed out by the spacing between residues in the 2nd paragraph of the paper. So, in figure 1, the packing of Type 2 helical barrels could be described as the i+4 ridge between residues e-1 and a (although the figure depicts a and e in the same heptad repeat, which would not be classical knob-into-holes packing), whereas the d and g packing are the i+3 ridge packing. Explaining the packing in this respect might help clarify the faces of the helix that are packing. Alternatively, the relative positions of the heptad repeat registers could be used. In the Type 2 packing, the registers are +4 shifted from each other.

Response 3 - Given that this is a complex phenomenon, we strive to have absolute clarity about this. The suggested classic work by Chothia, Levitt & Richardson (1981) describes ridges-into-grooves packing, which is a more general description of helix packing found mainly in globular proteins. We wish to avoid such descriptions, as the interactions described in our manuscript are explicitly coiled-coil knobs-into-holes interactions. We agree that the split interface description from our previous work

(Walshaw & Woolfson (2003)) can help understand the packing of Type-2 coiled coils. However, in the context of this manuscript we would be concerned that relabelling residues as suggested by Reviewer 1 might hamper understanding the packing of the lower symmetry structures. To improve clarity, however, we have incorporated the following sentences on page 3 of the Introduction.

“In these structures, two seams are presented on a single α helix allowing interaction with two neighbouring helices (Fig. 1f). These two seams can be considered as two overlaid Type-N interfaces (Fig. 1d). To avoid confusion, a single heptad repeat is described where one seam has knob residues at *a* and *e* and the other knob residues at *d* and *g*.”

*On a more minor note, I could not find the reason for listing the heptad repeat starting with the *g* position in Table 1 as well as what amino acids filled position *f*.*

Response 4 - The heptad repeats were described from *g* to *f* in Table 1 for continuity with our previous publication (Thomson, A. *et al.* 2014 **Science** 485-48). In light of Reviewer 1's comments, we realise that in the context of the new paper describing them from *a* to *g* is more logical and we have revised the manuscript accordingly. In addition, the following sentence has been added to the Table caption.

“Repeating *f* positions are occupied from N to C terminus by Q, K, W and Q respectively.”

*Lastly, Crick's original 1953 knobs-into-holes paper only referred to a knob repeating every 7 residues. It would be good to reference the first paper denoting the “abcdefg” for the heptamer repeat as well as the identification of the *a* & *d* residues as the knobs in the packing.*

Response 5 – The Reviewer is correct. We apologise for the oversight and have added the following references:

Cohen C. & Holmes K. C. X-ray diffraction evidence for alpha-helical coiled-coils in native muscle. *6*, 423-432 (1963).

Sodek J., Hodges R. S., Smillie L. B. & Jurasek L. Amino-Acid Sequence of Rabbit Skeletal Tropomyosin and Its Coiled Coil Structure. *P Natl Acad Sci USA* **69, 3800-3804 (1972).**

In summary, this work contributes fundamentally to understanding protein structure prediction. The analysis was performed systematically. Moreover, the wealth of supplementary data was presented in a very organized that is easily digested.

Response 6 - We are delighted by the Reviewer's summary and that s/he finds the supplementary information useful.

Reviewer #2

Rhys et al describe the design and characterization of several families of novel coiled coil sequences and structures. The new macromolecular assemblies are characterized in detail, both in solution and as crystal structures. The structures show that slight changes in sequence can yield completely different oligomeric states. Moreover, some of the assemblies form barrels with internal pores, while others form collapsed structures with little or no interior cavity. Interestingly, some of the structures are symmetric (as would be expected from homo-oligomers), while others

are (surprisingly) non-symmetric. The results are compelling and the take-home messages are interesting.

The paper is well-written and is a pleasure to read. An enormous amount of experimental work was done, and it is described clearly and concisely. Importantly, the manuscript does not merely list a bunch of sequences and structures. Instead, the authors do an excellent job explaining how/why subtle changes in sequence - sometimes involving just a few atoms - give rise to different packing and ultimately to dramatically different 3D structures.

This work demonstrates how far the field of de novo protein design has advanced in recent years.

Response 7 – We are pleased that Reviewer 2 has highlighted the novelty of homooligomers forming low-symmetry structures. To emphasise this, we have decided to alter the manuscript's title slightly to "Maintaining and breaking symmetry in **homomeric** coiled-coil assemblies". We are grateful for his/her positive comments on the manuscript in general.

Reviewer #3

The paper is a follow-up to an earlier work by the same authors on the design of α -helical barrels. Such structures have the KIH packing and are therefore classified as coiledcoils, albeit very unusual ones. The authors depart from their previous design CC-Hept (which gets a new name here). This structure is based on a 30-mer peptide with the heptad sequence (gabcdef) A?KE?A* , where "?" stands for the residues in a and d positions. The authors vary the latter systematically and analyze the resulting structures. This gives 9 possibilities, of which 3 could not be crystallized. Of the remaining 6 structures (some of which have been previously published) form four symmetrical α -helical barrels. The remaining two form asymmetric "collapsed" structures, i.e. without a pronounced channel in the middle. Some additional variations are also analyzed.

Based on this data, the authors suggest that symmetric α -helical barrels of this type are stabilized by β -branched residues in a and d positions. This prevents them from collapsing and forming the (logically expected) hydrophobic core. Some computational modeling is done, and seems to support the conclusions.

The work contains interesting new data and conclusions that advance the field (design of artificial α -helical barrels). The various asymmetric oligomers described are in fact unusual and rare and thus interesting. My most important remarks are as follows.

Response 8 – We are pleased by the Reviewers positive and supportive comments.

I found the paper quite difficult to read. This is partly due to the complexity of the phenomena studied, many of which can not be fully explained. However, a condensed style of writing as well as suboptimal preparation of display elements further aggravate the problem.

Response 9 – Given the very positive comments of Reviewers 1 & 2, in particular on how the manuscript and SI are written and presented, we are surprised by this comment from Reviewer 3. Nonetheless, as we wish to appeal to the broad

readership of *Nat Commun* and to present the work in the best possible way, we have tried to address Reviewer 3's comments as follows.

In particular, I believe that the figures 3 and 4 need to be carefully revised, as indeed crucial for the understanding. The reader is lost in all these multiple panels, some of which closely overlap in the message. The panels must also be rearranged in a more logical order (for example, a multipanel figure with symmetric structures, another one with collapsed structures, and another for special cases such as aromatics). At present it is extremely difficult to correlate the figures and Table 1.

Response 10 – We agree that figure 3 and 4 should be revised. As a result of the manuscript being transferred from a sister *Nature journal*, we have the opportunity to increase the number of figures now. To this end, two new figures have been made. The first (Figure 5) contains sequences from the beta-branched-class open barrels and the second (Figure 6) contains the aromatic sequences and 5H2L-I9L. In so doing, each figure has a more-focused theme and is better aligned with the order that they are referenced in the main text. Caption titles for each figure have been rewritten to give a more direct description of the content.

“Fig. 3: Diversity of structures formed by the CC-Type2 variants with aliphatic cores.”

“Fig. 4: Coiled-coil bundles containing β L or LL cores.”

“Fig. 5: α -Helical barrel structures formed by the $\beta\beta$ class.”

“Fig. 6: Coiled-coil bundles with aromatic cores and 5H2L_2.1-I9L.”

The references to figures in the main text have been updated to reflect the new figures. Taken together, we hope that these significant changes improve the readability of the manuscript and better facilitate the narrative of the paper.

The central Table 1 can be improved by adding the column describing the assembly type (symmetric/asymmetric, barrel/collapsed etc) and a column with the PDB code. The old and new names of some assemblies should also be included. Can the authors provide some quantification of how dense the cores of the individual collapsed structures are? Are these as dense as in globular proteins? Some panels (e.g. Fig. 3b, right; Fig. 4b) suggest the opposite, but it is impossible to tell from the figures alone.

Response 11 – There is limited space in the table for additional columns. However, we have tried to accommodate Reviewer 3's suggestions, which are good. We have included a description of the structures alongside the oligomeric state of the crystal structure. While there is a comprehensive list of the PDB accession codes available in the data availability section, we appreciate that a quick look-up table can be useful. Therefore, we have replaced the resolution of the structures with a column of PDB accession codes. The new names for the structures already exist in Table-1 in the form of the list of suffixes. To make this clear without reading the main text, and to include former names, we have introduced a footnote as follows

“²Sequences are described as CC-Type2 with a unique suffix. CC-Type2-VI, CC-Type2-LI and CC-Type2-LV have been described previously as AVKEIA, CC-Hept and ALKEVA, respectively.²⁶”

On a quantification of core packing in the collapsed structures: we have not done this specifically. However, as noted on page 7 of the manuscript, all of these new structures test positive in SOCKET, *i.e.* they have knobs-into-holes packing of side chains between helices and in the core. In addition, by inspection of these experimental structures there are no voids in the collapsed structures. Thus, the side-chain packing in these structures is closer and therefore denser than that found in globular proteins where helices tend to pack *via* ridges-into-grooves.

The first 1.5 pages of the introduction (including most of Figure 1) can be sacrificed. It makes little sense to provide this supercondensed overview of all possible coiledcoil structures. The introduction should rather focus on the construction of α -helical barrels. It should be explained which sequence pattern and why was used to come to the structure CC-Hex, the starting point of the current project.

Here I suggest adding an appropriate 3D / depth-rendered view showing the involvement of sidechains of all six (gabcde) positions in the barrel stabilization. The flat, schematic drawings like the panel 3b (and many similar) hardly help.

Shortening the introduction should provide room for more detail throughout the manuscript, which would save the reader the trouble of looking up every single reference before the logic can be understood!

Response 12 - In order to contextualise coiled coils for a broad readership we feel it necessary to explain both canonical and complex coiled coils. As Reviewer 3 comments, we have limited this to a condensed description so that the focus of the majority of the introduction is on symmetrical α -helical barrels. In our opinion removing this description would remove necessary background and context leading to complex coiled coils that is needed to follow the results and discussion. Complex coiled coils contain a mixture of interface types that are described individually in the first 1.5 pages. This section also emphasises the symmetrical nature of most self-associating coiled coils so that readers can appreciate the novelty of the low-symmetry structures that we have discovered. In summary, we believe that first 1.5 pages are essential and, therefore, should be retained.

The support obtained by computation is interesting. However, since the result (=experimental structures) is known, the risk of biased conclusions is large. To me, much of the argumentation provided falls short of being strictly causative. Some more mechanistic and detailed explanations, if available, could help. (I liked the 'Computational protein threading' (last section), but here the authors themselves warn on the limitations of the scoring used).

Response 13 - It is not clear if Reviewer 3 is referring to the replica-exchange molecular dynamics, the sequence-threading analysis, or both. It is true that biased conclusions can be drawn from computational data when seeking to corroborate experiments. We have done our level best not to fall into this trap.

For the constant-pH replica-exchange molecular-dynamics simulations, we used a meticulous and reproducible protocol (which can be provided as a protocol-exchange document) starting from the crystal structures of CC-Type2-LL-L17E and CC-Type2-IL-Sg-L17E. We compared the results from two different sets of parameters and monitored convergence of the protonation state sampling for the internal Glu residues as well as replica walk in pH space. This demonstrates the accurate sampling of both the protonation and conformational spaces along the pH ladder as well as the robustness of our calculations. For calculating pKa's, as described in the main text, we believe that treating buried acidic residues that are in proximity as

polyprotic acids is justifiable. Because the results from our simulations are solely based on the crystal structures of CC-Type2-LL-L17E and CC-Type2-IL-Sg-L17E obtained at a specific pH, the ability to correlate calculated pKa's with solution-state experimental results obtained over a large pH range is remarkable. Moreover, the main conclusions drawn from this section are based on macroscopic pKa's that are used to interpret macroscopic behaviour of the analytes during circular dichroism experiments, therefore avoiding overinterpretation of the computational data.

Regarding the sequence threading section, please see Response 1. In addition, we appreciate that the ultimate structure formed by self-associating peptides depends on several factors. In this paper, we have tried to delineate one factor, *i.e.* steric interactions. In order to quantify this, we used the sequence-threading analysis. We understand that this could result in overinterpretation of the data. In an attempt to mitigate this, we have included the total energy plots, in addition to the steric contribution, which demonstrates the significance of the steric contribution. To avoid biases that might be intrinsic to a particular force field, we have rescored the structures using two additional force fields; again, please see Response 1.

It is also not very clear how far the conclusions can be applied outside of the very specific (A?KE?A) pattern. The given single example beyond this pattern, that of the modified 5H2L structure, which also does not seem to completely collapse, is not too convincing.*

Response 14 – This is addressed by Response 2.

REVIEWERS' COMMENTS:

Reviewer #1 (Remarks to the Author):

The authors have addressed my concerns. Re-reading the manuscript, they have also addressed the concerns of the other 2 reviewers. I appreciate the careful consideration and work that the authors put in their responses.